# Advances in Drug Treatments for Companion Animal Obesity

**DOI:** 10.3390/biology13050335

**Published:** 2024-05-11

**Authors:** Helena D. Zomer, Paul S. Cooke

**Affiliations:** Department of Physiological Sciences, University of Florida, Gainesville, FL 32610, USA; helenazomer@ufl.edu

**Keywords:** canine, feline, GLP-1, GIP, weight loss, tirzepatide, semaglutide

## Abstract

**Simple Summary:**

Companion animal obesity is a major concern worldwide, leading to health issues and reduced lifespan. Current treatments have had limited success, but new possibilities are emerging from human medicine breakthroughs. This review discusses how GLP-1 receptor agonists like semaglutide and tirzepatide, designed for human diabetes treatment and recently approved by FDA as weight loss drugs, may be useful for future veterinary applications. These drugs were proven effective in rodents, primates, and humans and are promising for combating obesity in companion animals. Despite multiple health benefits described in humans, further studies are needed to assess their safety and effectiveness in veterinary medicine.

**Abstract:**

Companion animal obesity has emerged as a significant veterinary health concern globally, with escalating rates posing challenges for preventive and therapeutic interventions. Obesity not only leads to immediate health problems but also contributes to various comorbidities affecting animal well-being and longevity, with consequent emotional and financial burdens on owners. While past treatment strategies have shown limited success, recent breakthroughs in human medicine present new opportunities for addressing this complex issue in companion animals. Here, we discuss the potential of GLP-1 receptor agonists, specifically semaglutide and tirzepatide, already approved for human use, for addressing companion animal obesity. These drugs, originally developed to treat type 2 diabetes in humans and subsequently repurposed to treat obesity, have demonstrated remarkable weight loss effects in rodents, non-human primates and people. Additionally, newer drug combinations have shown even more promising results in clinical trials. Despite current cost and supply challenges, advancements in oral and/or extended-release formulations and increased production may make these drugs more accessible for veterinary use. Thus, these drugs may have utility in companion animal weight management, and future feasibility studies exploring their efficacy and safety in treating companion animal obesity are warranted.

## 1. Introduction

Companion animal obesity has reached alarming proportions in the United States, presenting a critical veterinary health concern [1,2]. According to a 2022 survey from the Association for Pet Obesity Prevention, 61% of cats and 59% of dogs in the U.S. are overweight or obese, representing a significant increase from previous years [3]. The escalating trends in U.S. pet obesity are part of a global epidemic, with studies covering Brazil, China, Russia, the United Kingdom and Spain showing comparably elevated rates [4,5]. These statistics collectively emphasize the desirability of addressing pet obesity through both preventive and therapeutic approaches.

The ongoing rise in companion animal obesity rates may involve genetics, diet and lifestyle habits [2,6]. Factors such as age, sex and surgical sterilization also contribute to obesity, emphasizing the multifactorial nature of the condition and its association with behavioral factors and feeding practices [2,6,7,8,9,10]. However, the single most important factor contributing to pet obesity is the owners’ underestimation of their pets’ body condition and misconception of what an animal at a healthy weight looks like [2]. Additionally, there is a clear association between owners’ overweight/obesity and the weight status of their pets, emphasizing the interplay between human and animal health [9,11,12]. This bidirectional impact of obesity in both human and animal health reinforces the One Health concept, which emphasizes the interdependence of the health of humans, animals and the environment and advocates for therapeutic strategies that consider the diverse factors influencing this epidemic.

Obesity not only poses immediate health risks including high blood pressure and breathing difficulties, but also contributes to the development of various comorbidities, such as type 2 diabetes, heart and liver diseases, osteoarthritis and cancer [13]. Obesity-related comorbidities contribute to diminished animal well-being, which imposes emotional and financial burdens on pet owners [1]. Additionally, obesity leads to profound consequences on animal longevity, and a negative correlation between obesity and lifespan has been demonstrated in both cats and dogs [7,14].

Current treatment of companion animal obesity involves a multifaceted approach with dietary management and behavioral modifications [4,15,16]. This includes a controlled calorie intake through balanced diets, and encouraging physical activity [12,17,18], underscoring the necessity of collaborative efforts between veterinarians and pet owners associated with comprehensive strategies to address the complexities of companion animal obesity. However, the alarming rates of obesity worldwide make it clear that present strategies have not been effective.

One reason for the failure of current pet obesity treatments is the lack of owner compliance [19,20]. Low compliance has been interpreted to result from an insufficient understanding of the importance of addressing pet obesity by the owners, especially when the benefits are unclear or long-term, and from the central role that feeding plays in emotional and social interactions between owners and their pets, especially dogs [1,19]. In communicating with owners, veterinarians must highlight not only the impact of obesity on overall health and lifespan but especially short-term health improvements and financial benefits to increase treatment adherence. In this sense, the most appealing reason from the owners’ perspective for the prevention and treatment of obesity may be to avoid the development of type 2 diabetes (T2D), a debilitating chronic disease commonly occurring in cats that is treated with expensive and laborious therapeutics [21]. Most patients with T2D will die prematurely within 2 years of their diagnosis due to the inability of owners to follow therapeutic protocols [21]. Recently, the assessment of glycosylated hemoglobin (HbA1c) has been optimized so that feline pre-diabetic conditions can be successfully diagnosed and treated before disease onset [22]. Therefore, pre-diabetic cats would especially benefit from weight loss in terms of significantly reducing their risk for T2D.

One weight loss drug, dirlotapide (Slentrol) has been approved to treat obesity in dogs. This drug, approved by the FDA in 2007, reduces appetite and also decreases fat absorption [23]. However, dirlotapide has not obtained widespread usage and only one paper has been published on this drug since its initial approval 17 years ago [23,24,25,26].

In contrast, several obesity drugs are available for human use, and a promising new generation of drugs has recently been approved by the FDA and adopted widely [27]. Why are these drugs already available to humans but these or similar drugs are not used in companion animals? The lack of weight loss drugs targeting pets may represent a substantial and overlooked opportunity for the veterinary market.

## 2. The Incretins: A Revolution in the Treatment of Human Type 2 Diabetes and Obesity

As is often the case, the origins of the current clinical treatments for obesity can be traced back to basic science discoveries that initially did not appear to have any clear relationship to body weight regulation or obesity. A timeline of key events leading to the development of the current GLP-1 receptor agonist weight loss drugs is shown in Figure 1. In 1964, the ability of intestinally administered glucose to evoke an insulin response was shown to be greater than that of a similar dose administered intravenously [28]. This suggested that some intestinal factor(s) could potentiate insulin secretion in response to glucose, confirming a hypothesis from decades earlier postulating the existence of an incretin, a gut hormone that stimulates insulin release from the endocrine pancreas [29]. The first of these incretin factors was identified in the 1970s (Figure 1), found to be produced in enteroendocrine K cells preferentially located in the upper intestine, and eventually named glucose-dependent insulinotropic polypeptide, or GIP [30,31]. Subsequent work showed that blocking GIP activity did not totally abolish the incretin effect [32], which led to the identification of glucagon-like peptide-1 (GLP-1) (Figure 1), which is produced in the L cells of the intestines, in 1987 [33].

Patients with T2D were shown to have an impaired incretin effect [36], and this is an important contributor to the pathologies associated with T2D. In a study that foreshadowed the subsequent use of incretins in T2D treatment, exogenous GLP-1 was shown to increase β-cell insulin secretion (Figure 2) to levels comparable to those in healthy control patients [37]. The demonstration that GLP-1 could promote satiety and decrease energy intake in humans [38] (Figure 2), along with other properties such as slowing gastric emptying and increasing insulin sensitivity (reviewed in [39]), emphasized that this hormone had other effects in addition to promoting insulin secretion. These effects on insulin secretion and sensitivity, appetite, satiety and gastric emptying appear to be the most critical facets of GLP-1 receptor agonist effects on body weight. However, in addition to these critical effects, GLP-1 receptor agonists have been shown to have a wide range of effects on organs and tissues all over the body, some of which have clear links to their weight-reducing abilities (Figure 2).

## 3. Semaglutide and Tirzepatide Effectively Reduce Body Weight

### 3.1. Semaglutide

The current use of GLP-1 receptor agonists in humans focuses on two drugs that have been recently approved for the treatment of both T2D and obesity, semaglutide and tirzapatide (Figure 1). Semaglutide, sold as Wegovy for obesity treatment and Ozempic for T2D treatment, is a long-lasting GLP-1 agonist. Semaglutide was originally developed for human T2D patients (at 1 mg/week, injected subcutaneously) but initial studies also showed concomitant body weight reductions [41]. Subsequent studies then evaluated higher doses of semaglutide in humans for effects on T2D and other parameters such as weight loss. A higher dose was shown to produce more pronounced beneficial effects on HbA1c as well as body weight in T2D patients [41,42].

Critically, in a study involving over 300 patients, semaglutide also reduced body weight in non-diabetic patients who were overweight or obese. Semaglutide treatment for 2 years (2.4 mg/day) reduced body weight by 15.2% in these patients (Table 1); the weight reduction in non-diabetic patients exceeded that in diabetic patients [43]. Side effects of this treatment were relatively mild, consistent with previous results with GLP-1 receptor agonists.

The ability of semaglutide to decrease HbA1c levels in diabetics clearly illustrates its beneficial effects on glycemic control in these patients. In addition to improved HbA1c, these patients also had positive changes in cholesterol, non-HDL cholesterol and blood pressure, parameters that are linked to cardiovascular risk, suggesting that semaglutide might improve cardiovascular outcomes in diabetic patients. A meta-analysis of cardiovascular risk in patients with T2D and cardiovascular disease indicated that GLP-1 receptor agonists reduced adverse cardiovascular events by 14% [49]. Semaglutide produced comparable cardiovascular benefits in patients with both T2D and cardiovascular disease or risk factors [50,51].

These results suggested that semaglutide might have significant cardiovascular benefits for obese patients without T2D. In a recent study of over 17,000 overweight and obese patients with pre-existing cardiovascular disease but without T2D, semaglutide treatment for a mean duration of 33 months reduced the risk of myocardial infarction, stroke or death from cardiovascular causes by 20% [52]. Of note, this longer treatment duration was not accompanied by an increased incidence or severity of side effects and a recently published meta-analysis of extended clinical trials (52 weeks) did not reveal any long-term deleterious effects associated with these drugs [53].

In addition to the demonstrated cardiovascular effects [52], recent results indicate that GLP-1 receptor agonists may have beneficial effects on systemic and gut inflammation in mice [54,55]. These studies showed that GLP-1 receptor agonists such as exenatide (which has been used experimentally in veterinary medicine) act to decrease systemic inflammation through actions in the central nervous system [54]. Inflammation is closely associated with aging, with anti-inflammatory agents increasing lifespan and pro-inflammatory agents producing decreases. Although the anti-inflammatory actions of GLP-1 were demonstrated in mice [54,55], similar effects may occur in humans and companion animals and the anti-inflammatory properties of GLP-1 receptor agonists could be important for some of their beneficial effects.

### 3.2. Tirzepatide

Tirzapatide, sold under the brand names Mounjaro and Zepbound for the treatment of T2D and obesity, respectively, is the second drug recently approved for T2D and obesity treatment (Figure 1). Tirzapatide is both a GLP-1 and GIP agonist.

Adults with T2D given daily doses of tirzapatide ranging from 5 to 15 mg/day over a period of 40 weeks had significant reductions in HbA1c, fasting serum glucose and body weight vs. placebo-treated controls [56,57]. These patients also showed dose-dependent reductions in body weight.

The effects on body weight suggested that tirzapadite might have similar effects in non-diabetic obese patients. Recent results involving over 2000 patients indicate that a 20-week dose-escalation period followed by a 52-week treatment with three tirzepatide doses (5, 10 and 15 mg) led to striking body weight reductions in obese patients (Table 1) [44]. Side effects of these treatments were mild and often transitory.

The highest tirzepatide dose produced a 20.9% decrease in body weight (Table 1). This body weight reduction was accompanied by the expected beneficial changes in various cardiovascular and metabolic parameters. Although the decreased body weight preferentially reflected loss of adipose tissue, some muscle tissue was also lost as shown by decreased lean mass following the weight reduction period. Although this is not optimal, similar results have been obtained in patients following lifestyle changes, suggesting that significant adipose mass decreases are associated with concomitant loss of muscle mass [58]. In addition to beneficial effects on adipose and metabolic parameters, recent results indicate that tirzepatide, like semaglutide, lowers the risks of atherosclerotic cardiovascular disease [59].

The striking effectiveness of tirzepatide and semaglutide for human weight loss has led to rapid increases in the use of these drugs. One important question that remains to be answered is the sustainability of weight loss following cessation of drug treatment. A recent study [60] administered a 36-week regimen of tirzepatide to obese patients or overweight patients with at least one other weight-related concern. Patients were then randomized into two groups, with one group receiving tirzepatide and one group placebo for an additional 52 weeks. Patients receiving tirzepatide for an additional 52 weeks showed a further 5.5% weight loss. Conversely, patients receiving the placebo for the final 52 weeks of the study showed a 14% weight gain vs. their weight at the end of the initial 36-week treatment period, although this group had a body weight almost 10% less than at the start of the study even at the end of the entire 88-week trial. Although regimens may be designed to minimize the weight regained following cessation of tirzepatide treatment, these results suggest that significant increases in body weight may occur following cessation of tirzepatide treatment. Whether this could lead to an extended or even life-long continuation of a maintenance dose of drugs such as tirzepatide is presently unclear. The sharp rebound in body weight of human patients following cessation of tirzepatide treatment suggests that if this approach to weight management was adopted in veterinary medicine, administering these drugs would likely not just be a matter of a relatively short-term treatment that would lead to the establishment and maintenance of more desirable weight. Indeed, usage of these types of drugs in companion animals might involve either long-term or even permanent treatment.

Although semaglutide and tirzapatide have ushered in a new era in the treatment of human obesity, the development of more effective second-generation therapies is likely. Some of the drugs or drug combinations currently in clinical trials are described below.

## 4. Drug Combinations in Current Testing

### 4.1. Retatrutide, a GLP-1, GIP and Glucagon Agonist

The striking benefits of drugs that act as GLP-1 receptor agonists or agonists for both GLP-1 and GIP have provided novel and powerful tools to treat obesity and T2D. Recent results have suggested that new drug combinations acting on other targets, in addition to the GLP-1/GIP receptors, may have even more beneficial effects in T2D as well as in weight control. Retatrutide is a drug that combines the GLP-1/GIP agonist activities of tirzepatide with additional agonistic effects on glucagon receptors. This drug has completed Phase 2 clinical trials for both type 2 diabetes and weight loss and is now being evaluated in larger Phase 3 studies [61]. Phase 2 results indicated marked clinical benefits for T2D [61] and obese [45] patients exceeding those seen with tirzepatide (Table 1).

In the study of retatrutide for T2D, participants given the highest dose showed an over 20% reduction in HbA1c at 36 weeks of treatment compared to the start of treatment [61]. These patients also had reduced body weight, triglycerides, non-HDL cholesterol and blood pressure. The HbA1c reductions obtained with retatrutide were comparable to those reported previously for semaglutide and tirzapatide [41,46].

Retatrutide may be more effective for weight loss in overweight and obese patients than previously studied drugs. To evaluate retatrutide effects on weight loss in obese patients, various doses of retatrutide were administered by weekly subcutaneous injection for 48 weeks. Patients in the highest dose group had body weight decreases of approximately 24% at the end of treatment and 26% of the patients receiving the highest dose (12 mg/wk) had weight reductions of 30% or more [45]. The greater effects of retatrutide compared to tirzepatide on body weight may reflect the beneficial effects of the glucagon agonist in retatrutide on energy expenditure [45] and this effect may contribute to the greater weight loss seen with the “triple agonist” action of retatrutide compared with tirzapatide. Based on the trajectory of the weight loss curves, it was suggested that body weight had likely not plateaued at the conclusion of treatment. In addition to body weight effects, patients taking retatrutide also showed beneficial changes in HbA1c, waist circumference, triglycerides, systolic and diastolic blood pressure and non-HDL cholesterol. As with tirzepatide or semaglutide, side effects were mild, suggesting that adverse reactions would be unlikely to preclude the use of these drugs in companion animals. These results suggest that glucagon receptor agonists, in conjunction with GLP-1 and GIP receptor agonists, may be more effective for weight loss than GLP-1/GIP agonists alone.

### 4.2. Servodutide (BI 456906), a GLP-1 and Glucagon Agonist

Another drug developed for weight loss in humans is servodutide, a dual GLP-1 and glucagon agonist. The glucagon/GLP-1 agonistic effect of servodutide is similar to retatrutide, but servodutide lacks the GIP agonist activity of retatrutide. Servodutide was effective in decreasing body weight and HbA1c in diabetic patients [62]. In a recent Phase 2 clinical trial, non-diabetic obese patients were given escalating doses for 20 weeks, followed by 26 weeks of a maintenance dose (Table 1) [47]. The highest dose of servodutide produced an approximately 15% weight loss, with side effects that were comparable to other GLP-1 receptor agonist drugs.

### 4.3. CagriSema (Semaglutide/Cagrilintide), a GLP-1/Amylin Agonist

Amylin, a pancreatic beta-cell hormone that is co-secreted with insulin, has effects on gastric emptying and satiety mimicking those of incretins [63]. Thus, amylin agonists would be expected to have beneficial effects on body weight and T2D when given alone [64] or in conjunction with semaglutide [46]. One such agonist is cagrilintide. A semaglutide/cagrilintide combination, dubbed CagriSema, produced decreases in body weight (15.6%) exceeding those seen with either semaglutide or cagrilintide alone in a 32-week Phase 2 clinical trial (Table 1) [46].

### 4.4. AMG 133 (Maridebart Cafraglutide), a GIP Antagonist and GLP-1 Agonist

Recent pre-clinical and Phase 1 clinical studies [48] have demonstrated the ability of AMG 133 (maridebart cafraglutide), a molecule with dual GIP receptor antagonist and GLP-1 receptor agonist activity, to produce weight loss and metabolic benefits in mice, monkeys and humans. Humans given 3 treatments with the highest dose of AMG 133 on days 1, 29 and 57 had a 14% reduction in body weight by approximately 3 months after the initiation of the study (Table 1). The AMG 133 had an extended half-life (about 20 days), which allowed longer intervals between doses. Notably, the body weight reduction seen at 3 months was only slightly reduced to 12% at day 210, about 5 months after the last AMG 133 dose.

The AMG 133 mechanism of action involving GIP antagonism with concomitant GLP-1 agonism initially seems paradoxical in light of the ability of tirzepatide, a dual GIP/GLP-1 agonist, to produce similar body weight decreases. However, the previous literature in experimental animals has supported the concept that inhibiting GIP signaling has beneficial effects on body weight [65,66] and reduced GIP receptor signaling is associated with decreased body mass index (BMI) in humans [67]. In addition, chronic GIP treatment has been reported to lead to downregulation of the GIP receptor and functional desensitization of the GIP receptor to its ligand [68]. Thus, AMG 133 and tirzepatide may both produce a functional inhibition of the GIP response and be similar in their mode of action.

### 4.5. Oral Formulations of Current Weight Loss Drugs

Current treatments of humans with tirzepatide and semaglutide require weekly self-injections, and all the drugs described in the section above that are now in development are also administered by injection. However, a number of pharmaceutical companies are currently developing second-generation obesity drugs that are efficacious and safe when taken orally, which is preferred by patients over the subcutaneous injections now used to administer these drugs. Initial Phase 1 and 2 clinical trials indicate that orally active forms of these types of drugs produce weight loss in humans comparable to that obtained with injected tirzepatide and semaglutide [69,70], suggesting they can be effectively administered orally. The availability of orally active versions of these drugs and the newer drugs in development would likely facilitate their use in animals such as dogs and cats, as it would avoid ongoing owner injections of the animals.

In summary, currently used GLP-1 drugs will likely be available soon in oral formulations that will facilitate the use of this class of drugs in both human and veterinary medicine. In addition, recent results suggest that despite the unprecedented weight loss seen with semaglutide or tirzepatide, combining these drugs with amylin or glucagon agonists may promote even more pronounced weight loss and clinical benefits. Despite the rapid progress in developing and testing weight loss drugs centered around the weight loss effects of GLP-1 receptor agonists, the progress to date is likely just the opening chapter in our understanding of a class of drugs that may now allow the effective treatment of obesity, a previously intractable problem in both human and veterinary medicine.

## 5. Can GLP-1/GIP Receptor Agonists Be Used for Weight Loss in Companion Animals?

### 5.1. Exenatide, a GLP-1 Agonist, Promotes Glycemic Control and Induces Weight Loss in Cats

Despite their striking efficacy and relatively mild side effects in humans, a critical question is whether the GLP-1 receptor agonists would be safe and effective in companion animals. Both human T2D and feline diabetes show a strong association with obesity, insulin resistance and impaired insulin secretion by β-cells [71]. The similarities in the pathophysiology of human and feline diabetes suggest that medications proven effective in human treatment would likely be effective in cats, and the same rationale may be extended to obesity drugs. However, studies on cats are limited. One study showed that liraglutide, a GLP-1 receptor agonist approved for human T2D and obesity, improved glycemic control during hyperglycemia in healthy cats by increasing insulin concentrations and decreasing glucagon concentrations. Both the pharmacokinetics and pharmacodynamics of this drug were similar to those reported in humans [72]. Liraglutide produced similar side effects in cats as the GLP-1 receptor agonist drugs have in people, such as nausea and vomiting, but in the cat study, the liraglutide dose was 15 times greater than the typical human dose [72].

A number of studies have shown the potential of exenatide in the treatment of T2D and weight loss. Exenatide is a synthetic analog of exendin-4, a peptide hormone discovered in 1992 in the saliva of Gila monsters (*Heloderma suspectum*) [73]. Exendin-4 is a natural ligand of the GLP-1 receptor and mimics GLP-1 actions. However, unlike GLP-1, which is rapidly metabolized within 2 min, exendin-4 has long-lasting activity, with a half-life of 2.4 h [74]. Exenatide was the first GLP-1 receptor agonist approved for clinical use in the United States (Byetta, Amylin Pharmaceuticals) in 2005 and since then has been extensively used for T2D treatment in humans and has also shown efficacy in diabetic cats [75,76,77].

Studies in both humans and cats have shown that exenatide improves glycemic control by reducing HbA1c levels and helps regulate blood glucose levels by enhancing insulin secretion and reducing glucagon levels, ultimately leading to better glucose utilization and reducing post-meal spikes [22,75,76,77,78]. Exenatide produced no reported side effects in cats, despite being used at doses that significantly exceeded that given to people [75,79,80]. Exenatide is also associated with significant weight loss in both humans and cats. In a large cohort study, human patients with T2D who received exenatide lost an average of 3 kg compared to those on insulin (who gained weight) [78]. Similar weight losses were seen in non-diabetic patients treated with exenatide [81].

The weight loss effect induced by exenatide is likely multifactorial, including appetite suppression by the delay in gastric emptying, leading to increased satiety and reduced food intake, and increased energy expenditure and lipolysis. Additionally, exenatide has potentially beneficial effects on cardiovascular risk biomarkers, as weight reduction with exenatide is associated with improvements in blood pressure and lipid profiles, which may contribute to a more favorable cardiovascular risk profile [78]. Findings in felines seem to reflect those observed in human trials, although it must be noted that most of the studies in cats tested the effects of exenatide in healthy animals, rather than diabetic and/or obese subjects [77].

Recent studies in cats have tested long-term extended-release technologies to minimize the burden of periodic injections of exenatide and improve pet owner treatment compliance [82,83]. Exenatide extended release is a microencapsulated formulation with a protracted pharmacokinetic profile that allows a once-weekly injection [76]. When tested in healthy cats [76], exenatide extended-release plasma concentrations peaked at 1 h and were still detected 4 weeks after a single subcutaneous injection (0.13 mg/kg). Additionally, fasting blood glucose and glucagon decreased, glucose tolerance improved and insulin concentrations increased in response to exenatide. Compared with preinjection values, continuous glucose monitoring showed that glucose concentrations decreased without inducing clinical hypoglycemia. In another study, a chemically controlled delivery system based on hydrogel microspheres was able to extend the half-life of a stable exenatide analog, [Gln^28^]exenatide, to 40 days after a single subcutaneous injection in cats [84]. Another strategy tested exenatide delivery through the subdermal implantation of an investigational delivery system, OKV-119. The device implantation procedure took only 1–2 min and resulted in measurable exenatide plasma concentrations for up to 35 days in healthy cats [82]. Further development of extended half-life strategies will facilitate the routine use of GLP-1 receptor agonists in veterinary clinical settings.

Exenatide has not been extensively used in dogs since diabetic dogs typically do not have significant beta cell populations remaining. This contrasts with cats, where drugs to improve insulin production/sensitivity by remaining beta cells are effective for a significant proportion of the diabetic population. Thus, diabetic dogs would typically not benefit from exenatide and it has not been used for treating canine diabetes. However, dogs have been used extensively in previous studies with exenatide and other GLP-1 receptor agonists [85,86]. Exenatide treatment in pre-diabetic dogs has been shown to increase insulin secretion, decrease food consumption and decrease body weight [85,87]. Exenatide and another GLP-1 receptor agonist did not have any deleterious effects on the dog pancreas [88] and led to improved glycemic control [89]. Critically, GLP-1 receptor agonists such as exenatide have been shown to be effectively absorbed in the canine digestive tract [89] and oral formulations of semaglutide and other GLP-1 receptor agonists are effective in dogs. Based on the previous literature, GLP-1 receptor agonists may provide a safe and effective means of reducing adiposity and body weight in dogs as well as cats, without extensive side effects. The two GLP-1 receptor agonist drugs currently in widespread use for human weight loss (semaglutide and tirzapatide) would likely produce weight loss and have extensive health benefits that parallel those presently described in rodents, non-human primates and humans, although optimal dosing regimens for dogs and cats, the amount of weight loss produced as a result and the side effects in these animals would have to be directly established.

### 5.2. Is Treatment with These Drugs a Practical Way to Treat Obesity in Companion Animals?

Even if drugs such as semaglutide and tirzapatide were shown to be safe and effective in companion animals, widespread adoption of this treatment approach for obesity would hinge on the economics of these types of therapies, which are more costly than the Slentrol (dirlotapide) currently used for canine obesity. Treatment of uninsured human patients with either tirzepatide or semaglutide presently costs over USD 1000/month [90]. This price point would likely severely reduce or eliminate a veterinary market for these drugs, as many owners might not be willing to spend this much on an ongoing basis to address their pet’s obesity. In addition, the intense medical and lay publicity regarding these drugs and their rapidly increasing usage for both T2D and obesity in human patients has led to current supply constraints. However, pharmaceutical companies are sharply increasing production to meet current demand, as well as in anticipation of an expanding market that may reach USD 100 billion/year by 2030. Furthermore, the development of additional GLP-1 products by other pharmaceutical companies may bring new sources of these types of drugs into the market. The smaller size of companion animals compared to humans would also suggest that the treatment of pets would require less of a drug if the effective doses were similar to those currently used in humans. Thus, despite cost and supply concerns that would currently make use in veterinary medicine problematic, the coming years will see an increased supply of these drugs. This makes it likely that the prices of these drugs will fall in coming years, which would facilitate their use in veterinary markets.

## 6. Conclusions

Companion animal obesity poses a significant global health challenge, necessitating innovative approaches for effective treatment. New treatments based on GLP-1 receptor agonists are revolutionizing human obesity therapies. This class of drugs may also present an unprecedented opportunity for veterinary medicine. Semaglutide and tirzepatide have demonstrated exceptional weight loss benefits in humans, and new drug formulations and delivery methods are being developed rapidly. Current cost and supply challenges will likely ease in coming years, and feasibility studies are warranted to explore the practicality and efficacy of these drugs in companion animal obesity management. Many hurdles must be overcome to develop these drugs for companion animals, such as identifying effective doses and dosing frequency, potential palatability issues for oral versions of these drugs and safety/side effect concerns. In humans, these drugs are most effective when given as part of a program involving exercise, dietary modifications and behavioral modifications. The same would likely be true for companion animals. However, the potential of GLP-1/GIP agonists may represent an unprecedented tool for addressing the complex issue of companion animal obesity.

## Figures and Tables

**Figure 1 biology-13-00335-f001:**
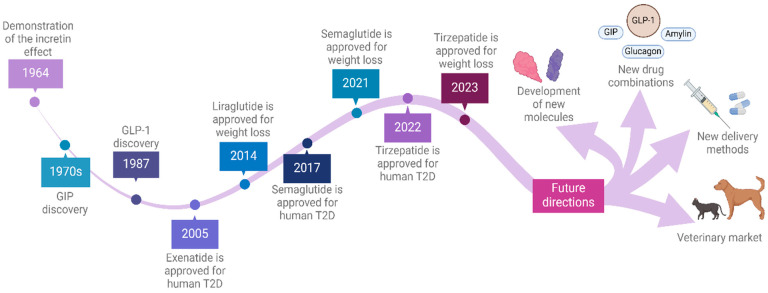
**Timeline of critical milestones in the development and use of GLP-1 receptor agonists.** The incretin effect, which indicated that a GI hormone(s) could increase insulin secretion from pancreatic β-cells, was demonstrated experimentally in 1964; the first hormone responsible for the incretin effect, GIP, was identified and sequenced in the early 1970s, and a second incretin, GLP-1, was identified in 1987. Exenatide, the first GLP-1-based drug, was approved for T2D treatment in humans in 2005. Another GLP-1-based drug, liraglutide, was approved by the FDA for the treatment of T2D under the brand name Victoza, then became the first GLP-1-based drug to be approved for weight loss (under the brand name Saxenda). Liraglutide produced modest weight loss benefits compared to semaglutide and tirzepatide; obese adults treated for 56 weeks with liraglutide had a body weight reduction of 5.6 kg (6% of body weight) more than placebo-treated controls [34]. Liraglutide was a commercial success, with 2023 sales of USD 2.65 billion, although this was far less than the sales of semaglutide (USD 17.8 billion) in the same year [35]. Semaglutide was approved for T2D in 2017 and subsequently achieved approval for weight loss 4 years later. Similarly, tirzepatide was initially approved for T2D in 2022, and then for weight loss the following year.

**Figure 2 biology-13-00335-f002:**
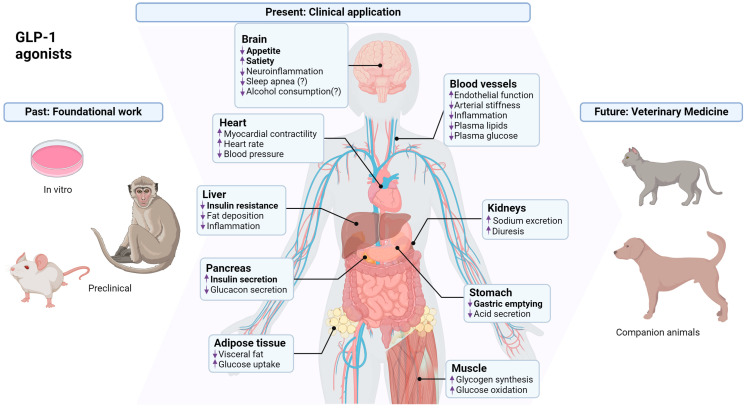
**Use of GLP-1 receptor agonists for the treatment of obesity: past, present and future.** GLP-1 receptor agonists were developed for the treatment of T2D and have been used clinically for this purpose in humans and experimentally in companion animals such as cats. These drugs produced weight loss in diabetic patients, leading to the development of weight loss drugs that were GLP-1 receptor agonists or GLP-1 receptor agonists with effects on other signaling pathways (GIP, glucagon, amylin). GLP-1 receptor agonists have pleiotropic effects both centrally and peripherally, although the bolded actions on insulin secretion and sensitivity, appetite, satiety and gastric emptying are most crucial for their body weight effects. GLP-1 receptors are best characterized in pancreatic beta cells and the brain, but the kidney, heart, GI tract and other organs also express this receptor [40]. Long-term effects of GLP-1 receptor agonists on weight loss reflect direct actions on target tissues, but also secondary effects of decreased body weight, which produce beneficial changes in blood pressure, insulin resistance, plasma lipids and other parameters. The putative effects of GLP-1 receptor agonists on sleep apnea have been reported by the drug manufacturer based on phase 3 clinical data but are currently unpublished. Similarly, clinical trials to evaluate the suggested beneficial effects of alcohol consumption are ongoing. Therefore, potential effects on these parameters are followed by a question mark to indicate that there is currently no published literature supporting these actions.

**Table 1 biology-13-00335-t001:** **GLP-1 based drugs currently used or in development for weight loss in human patients.** Two drugs are approved by the FDA and currently used for human weight loss (semaglutide and tirzepatide), while numerous other drugs have shown efficacy in phase 1 or 2 of clinical testing for weight loss and are currently undergoing additional human clinical trials.

Drug	Action	Clinical Trial	Maximal Weight Change	Dose and Administration	Ref.
**Semaglutide**	GLP-1 agonist	Phase 3	−15.2%	2.4 mg given by once-weekly subcutaneous injection for 2 years	[43]
**Tirzepatide**	GLP-1 and GIP agonist	Phase 3	−20.9%	5–15 mg given by once-weekly subcutaneous injection; 20-week dose-escalation period followed by 52-week treatment	[44]
**Retatrutide**	GLP-1, GIP and glucagon tri-agonist	Phase 2	−24.2%	1–12 mg given by once-weekly subcutaneous injection for 48 weeks	[45]
**Servodutide**	GLP-1 and glucagon agonist	Phase 2	−14.9%	0.6–4.8 mg by once-weekly subcutaneous injection; 20-week dose escalation period, followed by 26-week treatment	[46]
**CagriSema**	GLP-1 and amylin agonist	Phase 2	−15.6%	2.4 mg each of semaglutide and cagrilinitide by once-weekly subcutaneous injection for 32 weeks	[47]
**AMG 133**	GLP-1 agonist and GIP antagonist	Phase 1	−14.0%	140–420 mg on days 1, 29 and 57 body weight change at approximately day 90 shown	[48]

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
