# Peer review of "Advances in Drug Treatments for Companion Animal Obesity"

_biology, 2024, doi:10.3390/biology13050335_

Round 1

Reviewer 1 Report

Comments and Suggestions for Authors

The regulation of excess body weight in animals by modern analogues of GLP-1 is a practically important topic that has not yet received sufficient reflection. The authors quite successfully combined the depth of presentation and simplicity of form in the reviewed literature review. Probably now this review article, on the one hand, is simple enough to understand to attract the attention of researchers just beginning to study this area, and, on the other hand, is detailed enough not to be superficial. The authors presented in simple terms the current advances in the field of GLP-1.

Technical shortcomings of the manuscript:

1. References to literature should be in square brackets.

2. In Figure 2, human kidneys are shown incorrectly. You need to indicate their outline, but in the place where they should be located.

3. Lines 134-135: unclear wording.

4. Lines 137-142 shall be included in any subsection of the third section of this article. You cannot leave text fragments that do not relate to any subsection.

5. The column names in Table 1 are not visible. They may contain errors.

6. Line 158: error.

7. Line 181: per day or per week?

8. Lines 261-264: The sentence is very poorly worded.

9. The bibliography is not formatted according to the journal’s rules.

Author Response

Replies to Reviewers

The authors thank the reviewers for their positive and insightful comments, suggestions and criticisms regarding the manuscript, especially Reviewer #2 who pointed out a number of problems and concerns in the original version that have allowed us to significantly strengthen the revised version. We made extensive revisions and added a number of new references and information in response to all of the critiques.

Responses to Reviewer 1 

  1. References to literature should be in square brackets. Corrected as suggested.
  2. In Figure 2, human kidneys are shown incorrectly. You need to indicate their outline, but in the place where they should be located. We have corrected this by pointing to the left kidney, which is clearly shown in the diagram. We have also moved the line indicating the liver to the other side of the figure so everything will be clearer.
  3. Lines 134-135: unclear wording. Since the original submission of the manuscript, results of a phase 3 clinical trial on tirzapatide and sleep apnea have been reported by the manufacturer of this drug, but these results have not yet been published. We have re-written the last sentence of this paragraph to give this updated information and more clearly explain the question marks following sleep apnea and alcohol consumption in Figure 2.
  4. Lines 137-142 shall be included in any subsection of the third section of this article. You cannot leave text fragments that do not relate to any subsection. Corrected as suggested. Thank you for this suggestion.
  5. The column names in Table 1 are not visible. They may contain errors. Corrected as suggested.
  6. Line 158: error. Sentence rewritten.
  7. Line 181: per day or per week? Sentence rewritten to clarify dosing regimen.
  8. Lines 261-264: The sentence is very poorly worded. This sentence has been broken into 2 and rewritten to make it clearer.
  9. The bibliography is not formatted according to the journal’s rules. Corrected as suggested.

Reviewer 2 Report

Comments and Suggestions for Authors

Review Report of Manuscript ID Number “Biology-2963125”

The review article entitled “Advances in drug treatments for companion animal obesity” submitted for publication in the journal “Biology ” with manuscript ID Number “biology-2963125” seems quite interesting topic and could be attractive to the researchers all around globe. The study presents an intriguing perspective on the potential use of GLP-1 agonists, specifically semaglutide and tirzepatide, in addressing companion animal obesity. While the topic is timely and relevant given the increasing rates of obesity in companion animals, there are several aspects of the study that warrant critical examination:

The study revealed that these drugs were originally developed for the treatment of type 2 diabetes in humans and subsequently repurposed for obesity. Although the results in rodents, non-human primates, and humans are promising, there's no direct evidence presented regarding their efficacy and safety in companion animals as they have different metabolic and physiological responses compared to humans or other animals, which could influence the drug's effectiveness and safety.

This review article lacks long-term studies or potential side effects over extended periods. Long-term safety and efficacy data are crucial, especially when considering chronic conditions like obesity in companion animals.

The cost-effectiveness of these drugs compared to existing treatments or interventions for companion animal obesity need to be evaluated.

The practical aspects of administering these drugs to companion animals, such as dosing, frequency, and potential palatability issues, needs to be addressed.

 The study lacks empirical data supporting the use of GLP-1 agonists in companion animals for obesity management.

Considering the broader context of companion animal obesity, a multifaceted approach involving diet, exercise, and behavioral interventions remains crucial for effective prevention and management.

The present study represents the course of evolution of the Incretins but did not present their mechanism of action. My key suggestion is to present a simplified diagram of Mechanism of action of Incertins along with various factors.  It will help in understanding reader the mist of study about Incertins and their basic mechanism and will accurately help in understanding their employment on companion animals. Lines 86-95 are unable to explain the mechanism of actions of Incertins.

In Figure 1, the timeline doesn't mention other key developments in the field of incretin biology and GLP-1 research during this period, which could provide a more comprehensive historical context.

The timeline also stated that GLP-1 was identified in the late 1980s, which is accurate. However, it would be beneficial to specify the exact year (1987) and provide a brief description of the research or studies that led to its identification.

While the timeline mentions the approval dates for Exenatide, liraglutide, semaglutide, and tirzepatide, it lacks details on the clinical trials, efficacy, safety data, and regulatory processes that led to these approvals.

The timeline mentions that liraglutide produced "modest weight loss benefits" compared to semaglutide and tirzepatide. However, it doesn't provide specific data or references to support this claim. Including comparative efficacy data from clinical trials or studies would strengthen this assertion.

The statement that liraglutide "never achieved the commercial success of these later GLP-1 based drugs" lacks context and supporting data. While semaglutide and tirzepatide may have gained more attention or market share in recent years, it's essential to provide data or references to substantiate this claim regarding commercial success.

The timeline indicates that semaglutide achieved approval for weight loss 4 years after its approval for T2D and tirzepatide was approved for weight loss the following year. While this provides a general timeline, it would be more informative to include specific details on the clinical trials, patient populations studied, weight loss outcomes, and safety profiles leading to these approvals.

On line 144 “Semaglutide was originally developed for T2D (at a 1 mg dose given weekly by sub- cutaneous injection) but initial studies also showed beneficial effects on body weight (34)”. Did not mention the nature of clinical trial either acute or chronic experimentation or the organism either mammalian/human model.

The main title headings of Table 1 are missing or unable to visualize; kindly make it visible if so.

In Table 1  the line Two drugs have  been approved (by which authority..FDA ? or other?)

Table 1 does not present any side effects or toxic effects body weight ratio or BWI (body weight index) or the complexities of drug use. The recommendation authority in each case of drug should be mentioned for each phase and also it lacks recommendation body for its employment on companion animals.

Table 1 or its description below does not explain the high weight loss effect of Retatrutide even at its phase 2 trials.

The Lines 172-173 states “In addition to the demonstrated cardiovascular effects, recent results indicate that GLP-1 agonists may have beneficial effects on systemic inflammation in mice (41)” but no weight loss or related experimentation….please elaborate with reference to obesity and other references.

On line 177-178, similar effects may occur in humans and companion animals and GLP-1 agonists could produce significant life-extension through this mechanism.” No reference studies for this study and results.

Lines 222-223 “these results are somewhat predictable and unlikely to compromise the promise of these drugs for treating companion animals.” These lines are without any reference and recommendation trials. It’s just based on mechanistic approach without any experimentation.

Lines 253-254 “These results suggest that glucagon receptor agonists, in conjunction with GLP-1 and GIP 253 receptor agonists, may be more effective for weight loss than GLP-1/GIP agonists alone.”

This statement is given without any mechanistic approach or diagram. How could the synergistic effects of GLP-1 and GIP 253 receptor agonists impart positive impact in T2D/obesity treatment?

Also the whole story does not match support the evidence for treatment of companion cell obesity as different physiological conditions and other factors variable for each mammalian model including dose, frequency and time duration.

Lines 295-298 “Initial Phase 1 and 2 clinical trials indicate that orally active forms of these types of drugs produce weight loss in humans comparable to that obtained with injected tirzepatide and semaglutide (60,61) . This suggests that soon these drugs or their successors will be administered predominately through oral dosage.” These lines present the benefits of oral dosage over injections but did not present any reason for the oral effectiveness of drug compared to injected forms. References should be added or logical reasoning should be added to support this statement.

On line 314-316 the author claimed “The similarities in the pathophysiology of human and feline diabetes suggest that medications proven effective in human treatment would likely be effective in cats, and the same rationale may be extended to obesity drugs. However, studies in cats are limited.” How the author could impart the effect of limited or few studies on the overall treatment of obesity and T2D in cats? On one side he is recommending it a valid treatment but on other side he is accepting its limited research.

On line 320 the author claimed “The pharmacokinetics of this drug were similar to that reported 320 in the human (63).” What about other factors such as pharmacodynamics?  And what about its ADME i.e. absorption, distribution, metabolism, and excretion?

On line344-345 the author claim “Although findings in felines seem to reflect those observed in human trials, most of the studies in cats tested the effects of exenatide in healthy animals, rather than diabetic and/or obese subjects (68). “ so the statement or recommendation become ambiguous as feline model is concerned. Please remove such ambiguities in the whole manuscript. On one part the author is supporting the treatment on mechanistic basis but not presenting its studies/references to support his answers and also accepting the lack of experimental evidence which making it confusing and complicated”

The lines 350-355 “When tested in  healthy cats, exenatide extended-release plasma concentrations peaked at 1 h and were  still detected 4 weeks after a single subcutaneous injection (0.13 mg/kg). Additionally, fasting blood glucose and glucagon decreased, glucose tolerance improved and insulin concentrations increased in response to exenatide. Compared with preinjection values, continuous glucose monitoring showed that glucose concentrations decreased without in ducing clinical hypoglycemia.” Lack reference.?

On lines 375-378 “Based on previous literature, GLP-1 agonists may provide a safe and effective means of reducing adiposity and body weight in dogs as well as cats, and these drugs would likely have extensive health benefits that parallel those presently described in rodents, cats, non-human primates and humans.” The author recommends mechanistic actions of these drugs on all mammalian model including dogs, cats, mice etc. but did not narrated other factors those may affect the efficiency of these drugs and could cause side effects due to nature of animal and their physiochemical conditions.

On line 383-384 “This price point would likely severely reduce or eliminate a veterinary market for these drugs.” How and why?

On line 395-396 “Thus, it is likely that they could be produced in a cost-effective manner that would allow them to be used in veterinary markets in coming years.” If demand is more and supply would be short then it could inverse the situation so this reason is not enough to make them cost effective. The other reason could be the large production and ease in raw material and more licencing in pharmaceutical manufacturing could be more effective and suitable ways to cut doe the price.

The conclusion needs to be further refined and brief for the validation of these drugs on companion animals. Please revise it with brief and precise findings.

The overall language of the manuscript is good and does not require rephrasing except some systematics. Some sections of the review has extracted from others as it is and therefore around 20 % similarity index/plagiarism has been determined by turnitin while AI is too low which is a healthy sign about the originality of the manuscript. 

The review may not cover all relevant studies or may have a limited scope, potentially overlooking important research findings or alternative perspectives on the topic.

Addressing these limitations through rigorous study design, addition of diagrams/figures or systematic or graphical models, and careful interpretation of key results can enhance the validity and impact of the research findings.

Comments on the Quality of English Language

The overall language of the manuscript is good and does not require rephrasing except some systematics arrangements. Some sections of the review has extracted from others as it is and therefore around 20 % similarity index/plagiarism has been determined by turnitin while AI is too low which is a healthy sign about the originality of the manuscript. 

Author Response

Replies to Reviewers

The authors thank the reviewers for their positive and insightful comments, suggestions and criticisms regarding the manuscript, especially Reviewer #2 who pointed out a number of problems and concerns in the original version that have allowed us to significantly strengthen the revised version. We made extensive revisions and added a number of new references and information in response to all of the critiques.

Responses to Reviewer 2 

The study revealed that these drugs were originally developed for the treatment of type 2 diabetes in humans and subsequently repurposed for obesity. Although the results in rodents, non-human primates, and humans are promising, there's no direct evidence presented regarding their efficacy and safety in companion animals as they have different metabolic and physiological responses compared to humans or other animals, which could influence the drug's effectiveness and safety. Athough there is not a literature related to the use of semaglutide and tirzapatide in companion animals, there is a fairly extensive literature on the use of FDA-approved GLP-1 agonists (liraglutide and exenatide), especially in cats. We have enlarged these sections to more clearly discuss the effectiveness and safety of the GLP-1 drugs in companion animals.

This review article lacks long-term studies or potential side effects over extended periods. Long-term safety and efficacy data are crucial, especially when considering chronic conditions like obesity in companion animals. We agree that this is an important point that should have been discussed more thoroughly. The best study related to this is the one related to cardiovascular risk following semaglutide; in this study, 17,000 people were followed during almost 3 years of semaglutide. We have added a sentence in the revision that indicates that this patient population had no increases in side effect incidence or severity following this extended treatment.

The cost-effectiveness of these drugs compared to existing treatments or interventions for companion animal obesity need to be evaluated. We have added a mention that the GLP-1 agonists are more expensive than the dirlotapide (Slentrol) treatment presently available for dogs (which costs only about $2/day). However, Slentrol is not widely used despite having been on the market for over 16 years, indicating that other factors than cost have kept this drug from being widely used.

The practical aspects of administering these drugs to companion animals, such as dosing, frequency, and potential palatability issues, needs to be addressed. In the revised Conclusion section, we have added a new sentence indicating that: “Many hurdles must be overcome to develop these drugs for companion animals, such as identifying effective doses and dosing frequency, potential palatability issues for oral versions of these drugs and safety/side effect concerns”.

 The study lacks empirical data supporting the use of GLP-1 agonists in companion animals for obesity management. We have added additional data and references to the revised text that describe the efficacy and tolerability of GLP-1 agonists (liraglutide, exenatide) in companion animals.

Considering the broader context of companion animal obesity, a multifaceted approach involving diet, exercise, and behavioral interventions remains crucial for effective prevention and management. We totally agree, and at the end of the revised Conclusion section we have added a new sentence mentioning these important points.

The present study represents the course of evolution of the Incretins but did not present their mechanism of action. My key suggestion is to present a simplified diagram of Mechanism of action of Incertins along with various factors.  It will help in understanding reader the mist of study about Incertins and their basic mechanism and will accurately help in understanding their employment on companion animals. Lines 86-95 are unable to explain the mechanism of actions of Incertins.  We already show the actions of GLP-1 agonists in Figure 2, so another diagram showing their mechanism of action seems redundant. We do agree though that we want to make it clear to the reader how these compounds work, and so in the text paragraph describing the actions of GLP-1 agonists and in the legend for Figure 2, we have emphasized that their actions insulin secretion and sensitivity, satiety, appetite and gastric emptying are most crucial for their body weight effects.

In Figure 1, the timeline doesn't mention other key developments in the field of incretin biology and GLP-1 research during this period, which could provide a more comprehensive historical context. We share the goal of making Figure 1 as informative as possible. After significant discussion of possible additions to Figure 1, we were concerned that adding more information to this figure might make it seen too “busy” and actually detract from its usefulness. Therefore, we have not made any changes to Figure 1.

The timeline also stated that GLP-1 was identified in the late 1980s, which is accurate. However, it would be beneficial to specify the exact year (1987) and provide a brief description of the research or studies that led to its identification. Changed as suggested.

While the timeline mentions the approval dates for Exenatide, liraglutide, semaglutide, and tirzepatide, it lacks details on the clinical trials, efficacy, safety data, and regulatory processes that led to these approvals.  We have added more information on the efficacy and side effects of exenatide and liraglutide in companion animals, along with two new references (cats). The two most critical trials for showing the utility of GLP-1 agonists in humans were the trials of semaglutide and tirzapatide in non-diabetic humans. We have added the patient numbers for both of these trials. For the tirzapatide trial, the patient population was over 2,000 individuals, so we appreciate the suggestion to add this type of information because it clearly shows that conclusions reached were based on a sizable patient group.

The timeline mentions that liraglutide produced "modest weight loss benefits" compared to semaglutide and tirzepatide. However, it doesn't provide specific data or references to support this claim. Including comparative efficacy data from clinical trials or studies would strengthen this assertion. We have added new data and a reference to indicate that obese adults treated for 56 weeks with liraglutide had a body weight reduction of 5.6 kg (6% of body weight) more than placebo-treated controls.

The statement that liraglutide "never achieved the commercial success of these later GLP-1 based drugs" lacks context and supporting data. While semaglutide and tirzepatide may have gained more attention or market share in recent years, it's essential to provide data or references to substantiate this claim regarding commercial success. We have changed this sentence and added new data and a reference to indicate that in 2023, liraglutide (sold by Novo as Saxenda for obesity and Victoza for type 2 diabetes) had sales of $2.65 billion, as compared to $17.8 billion for semaglutide.

The timeline indicates that semaglutide achieved approval for weight loss 4 years after its approval for T2D and tirzepatide was approved for weight loss the following year. While this provides a general timeline, it would be more informative to include specific details on the clinical trials, patient populations studied, weight loss outcomes, and safety profiles leading to these approvals. As indicated above, we have added the patient numbers and information about the side effects observed for both the critical semaglutide and tirzapatide trials that led to their approval by FDA.

On line 144 “Semaglutide was originally developed for T2D (at a 1 mg dose given weekly by sub- cutaneous injection) but initial studies also showed beneficial effects on body weight (34)”. Did not mention the nature of clinical trial either acute or chronic experimentation or the organism either mammalian/human model. In the revised text, we have clarified that these studies are with human patients.

The main title headings of Table 1 are missing or unable to visualize; kindly make it visible if so. Changed as suggested.

In Table 1  the line Two drugs have  been approved (by which authority..FDA ? or other?) FDA. We have added this information to the Table 1 legend.

Table 1 does not present any side effects or toxic effects body weight ratio or BWI (body weight index) or the complexities of drug use. The recommendation authority in each case of drug should be mentioned for each phase and also it lacks recommendation body for its employment on companion animals. We have indicated that two drugs in this table are FDA-approved for human weight loss, and that all of the trials described in that table are human clinical trials. The GLP-1 based drugs that have been used and approved in companion animals are discussed elsewhere in the review. 

Table 1 or its description below does not explain the high weight loss effect of Retatrutide even at its phase 2 trials. We have added a short explanation of how the “triple agonist” actions of retatrutide could act to promote greater weight loss than was seen with tirzapatide.

The Lines 172-173 states “In addition to the demonstrated cardiovascular effects, recent results indicate that GLP-1 agonists may have beneficial effects on systemic inflammation in mice (41)” but no weight loss or related experimentation….please elaborate with reference to obesity and other references. We have added the refence for the cardiovascular data, which was discussed in the previous paragraph. We have also added one additional reference showing anti-inflammatory properties of GLP-1 agonists, and also indicated that GLP-1 agonists inhibit both gut and systemic inflammation. The inflammation studies were basic science studies in mice, and thus did not look at obesity, weight loss, etc. effects of GLP-1 agonists.

On line 177-178, similar effects may occur in humans and companion animals and GLP-1 agonists could produce significant life-extension through this mechanism.” No reference studies for this study and results. We have also added the one new reference showing anti-inflammatory properties of GLP-1 agonists. Our original statement that this anti-inflammatory effect could be linked with the longevity effects of GLP-1 agonists was probably too strong, and we have revised this to just indicate that the anti-inflammatory effects could be involved with some of the beneficial effects of these drugs.

Lines 222-223 “these results are somewhat predictable and unlikely to compromise the promise of these drugs for treating companion animals.” These lines are without any reference and recommendation trials. It’s just based on mechanistic approach without any experimentation. We have rewritten this section to clarify it and indicate that treatment with these drugs in companion animals would potentially be a long term or even permanent proposition.

Lines 253-254 “These results suggest that glucagon receptor agonists, in conjunction with GLP-1 and GIP receptor agonists, may be more effective for weight loss than GLP-1/GIP agonists alone.” This statement is given without any mechanistic approach or diagram. How could the synergistic effects of GLP-1 and GIP receptor agonists impart positive impact in T2D/obesity treatment? This section has been re-written and clarified.

Also the whole story does not match support the evidence for treatment of companion cell obesity as different physiological conditions and other factors variable for each mammalian model including dose, frequency and time duration. We agree, and we now emphasize that things like dose, frequency, etc. would have to be worked out in companion animals before these drugs could be efficaciously used in these species.

Lines 295-298 “Initial Phase 1 and 2 clinical trials indicate that orally active forms of these types of drugs produce weight loss in humans comparable to that obtained with injected tirzepatide and semaglutide (60,61) . This suggests that soon these drugs or their successors will be administered predominately through oral dosage.” These lines present the benefits of oral dosage over injections but did not present any reason for the oral effectiveness of drug compared to injected forms. References should be added or logical reasoning should be added to support this statement. We have added more information about what is required in order to give a drug orally, as well as why this is a better route of administration.

On line 314-316 the author claimed “The similarities in the pathophysiology of human and feline diabetes suggest that medications proven effective in human treatment would likely be effective in cats, and the same rationale may be extended to obesity drugs. However, studies in cats are limited.” How the author could impart the effect of limited or few studies on the overall treatment of obesity and T2D in cats? On one side he is recommending it a valid treatment but on other side he is accepting its limited research. As indicated above, we have added additional information to the revised text that describes the efficacy, tolerability and side effects of GLP-1 agonists (liraglutide, exenatide) in companion animals, as well as their ability to cause weight loss in these animals.

On line 320 the author claimed “The pharmacokinetics of this drug were similar to that reported in the human (63).” What about other factors such as pharmacodynamics?  And what about its ADME i.e. absorption, distribution, metabolism, and excretion? Both the pharmacokinetics and pharmacodynamics of the drug were similar to what has been observed in humans, and we have changed the text to indicate this.

On line344-345 the author claim “Although findings in felines seem to reflect those observed in human trials, most of the studies in cats tested the effects of exenatide in healthy animals, rather than diabetic and/or obese subjects (68). “ so the statement or recommendation become ambiguous as feline model is concerned. Please remove such ambiguities in the whole manuscript. On one part the author is supporting the treatment on mechanistic basis but not presenting its studies/references to support his answers and also accepting the lack of experimental evidence which making it confusing and complicated”  We have rewritten this section to clarify it.

The lines 350-355 “When tested in  healthy cats, exenatide extended-release plasma concentrations peaked at 1 h and were  still detected 4 weeks after a single subcutaneous injection (0.13 mg/kg). Additionally, fasting blood glucose and glucagon decreased, glucose tolerance improved and insulin concentrations increased in response to exenatide. Compared with preinjection values, continuous glucose monitoring showed that glucose concentrations decreased without inducing clinical hypoglycemia.” Lack reference? We have added this reference (Rudinsky et al, 2015).

On lines 375-378 “Based on previous literature, GLP-1 agonists may provide a safe and effective means of reducing adiposity and body weight in dogs as well as cats, and these drugs would likely have extensive health benefits that parallel those presently described in rodents, cats, non-human primates and humans.” The author recommends mechanistic actions of these drugs on all mammalian model including dogs, cats, mice etc. but did not narrated other factors those may affect the efficiency of these drugs and could cause side effects due to nature of animal and their physiochemical conditions. As indicated above, we have added additional information and references to the revised text that describes the efficacy, tolerability and side effects of GLP-1 agonists (liraglutide, exenatide) in companion animals, as well as their ability to cause weight loss in these animals.

On line 383-384 “This price point would likely severely reduce or eliminate a veterinary market for these drugs.” How and why? This sentence has been re-written to clarify why the current price of these drugs might negatively impact a veterinary application for these drugs.

On line 395-396 “Thus, it is likely that they could be produced in a cost-effective manner that would allow them to be used in veterinary markets in coming years.” If demand is more and supply would be short then it could inverse the situation so this reason is not enough to make them cost effective. The other reason could be the large production and ease in raw material and more licensing in pharmaceutical manufacturing could be more effective and suitable ways to cut down the price. This sentence has been re-written and clarified.

The conclusion needs to be further refined and brief for the validation of these drugs on companion animals. Please revise it with brief and precise findings. We have condensed the previous conclusion substantially, although a couple of new points suggested by the referees have also been added to this section.

The review may not cover all relevant studies or may have a limited scope, potentially overlooking important research findings or alternative perspectives on the topic. Addressing these limitations through rigorous study design, addition of diagrams/figures or systematic or graphical models, and careful interpretation of key results can enhance the validity and impact of the research findings. We hope that the new data, discussions and references that we have added in the revised version have addressed these concerns.

Reviewer 3 Report

Comments and Suggestions for Authors

Summary:

This manuscript discussed the potential of GLP-1 agonists, semaglutide and tirzepatide, for treating obesity in companion animals. The drugs, although initially targeting human type 2 diabetes, showed significant weight loss in different species. Additionally, the drug combinations are promising in clinical trials. Ongoing improvements in drug formulations and production methods could enhance their availability for veterinary applications. Therefore, these drugs potentially treat companion animals’ obesity, and future studies are focused on verifying their effectiveness and safety.

Major Comments:

1.      The manuscript discussed the advances in treating companion animal obesity using drugs, but most of the information was about the drugs used for humans. Humans and companion animals are different, and please provide information on the specific side effects of these drugs on animals.

2.      The subheadings of the manuscript need to be better connected. I recommend adding some sentences that link the different aspects.

3.      The manuscript mainly focused on GLP-1. However, there are other drugs available for treating companion animal obesity. Please discuss other drugs for companion animals’ obesity; for example, there are drugs that can inhibit microsomal triglyceride transfer protein.

Minor Comments:

1.      The first row of Table 1 is unclear; I recommend changing the font color of the first row.

2.      Please check the capitalization of the text in Figure 2, for instance, “Stiffness”.

Comments on the Quality of English Language

Minor editing of English language required.

Author Response

Replies to Reviewers

The authors thank the reviewers for their positive and insightful comments, suggestions and criticisms regarding the manuscript, especially Reviewer #2 who pointed out a number of problems and concerns in the original version that have allowed us to significantly strengthen the revised version. We made extensive revisions and added a number of new references and information in response to all of the critiques.

Responses to Reviewer 3 

Major Comments:

  1. The manuscript discussed the advances in treating companion animal obesity using drugs, but most of the information was about the drugs used for humans. Humans and companion animals are different, and please provide information on the specific side effects of these drugs on animals. Reviewer #2 also had this same concern. We have added new information to the revision on the types and severity of side effects seen with GLP-1 agonists (liraglutide and exenatide) in companion animals.
  2. The subheadings of the manuscript need to be better connected. I recommend adding some sentences that link the different aspects.  We have changed the manuscript in several places to better link the different subsections together.
  3. The manuscript mainly focused on GLP-1. However, there are other drugs available for treating companion animal obesity. Please discuss other drugs for companion animals’ obesity; for example, there are drugs that can inhibit microsomal triglyceride transfer protein. We have added new information to the revision at the end of the Introduction about dirlotapide (Slentrol), a drug that was approved by the FDA in 2007 to treat obesity in dogs that works by reducing appetite and decreasing fat absorption.

Minor Comments:

  1. The first row of Table 1 is unclear; I recommend changing the font color of the first row. Corrected as suggested.
  2. Please check the capitalization of the text in Figure 2, for instance, “Stiffness”. Corrected as suggested. 

Round 2

Reviewer 2 Report

Comments and Suggestions for Authors

The author has addressed all the issues and enormously improved the manuscript. I am satisfied with the efforts of author and all modifications have been made to make it suitable for publication, therefore i recommend its accepetance and subsequent publication in the journal of Biology.

Reviewer 3 Report

Comments and Suggestions for Authors

No further comments.

Comments on the Quality of English Language

Minor editing of English language required